# Preparation and Characterization of Self-Assembled Poly(l-Lactide) on the Surface of β-Tricalcium Diphosphate(V) for Bone Tissue Theranostics

**DOI:** 10.3390/nano10020331

**Published:** 2020-02-15

**Authors:** Jan A. Zienkiewicz, Adam Strzep, Dawid Jedrzkiewicz, Nicole Nowak, Justyna Rewak-Soroczynska, Adam Watras, Jolanta Ejfler, Rafal J. Wiglusz

**Affiliations:** 1Institute of Low Temperature and Structure Research, Polish Academy of Sciences, Okolna 2 str., 50-422 Wroclaw, Poland; j.zienkiewicz@intibs.pl (J.A.Z.); a.strzep@intibs.pl (A.S.); n.nowak@intibs.pl (N.N.); j.rewak@intibs.pl (J.R.-S.); a.watras@intibs.pl (A.W.); 2Department of Chemistry, University of Wroclaw, F. Joliot-Curie 14 str., 50-383 Wroclaw, Poland; dawid.jedrzkiewicz@chem.uni.wroc.pl (D.J.); jolanta.ejfler@chem.uni.wroc.pl (J.E.); 3Centre for Advanced Materials and Smart Structures, Polish Academy of Sciences, Okolna 2 str., 50-950 Wroclaw, Poland

**Keywords:** β-tricalcium diphosphate(V), Ce^3+^ and Pr^3+^ ions co-doping, poly(l-lactide), theranostics

## Abstract

This work was aimed to obtain and characterize the well-defined biocomposites based on β-tricalcium diphosphate(V) (β-TCP) co-doped with Ce^3+^ and Pr^3+^ ions modified by poly(l-lactide) (PLLA) with precise tailored chain length and different phosphate to polymer ratio. The composites as well as β-tricalcium diphosphate(V) were spectroscopically characterized using emission spectroscopy and luminescence kinetics. Morphological and structural properties were studied using X-Ray Diffraction (XRD) and Scanning Electron Microscopy (SEM). The self-assembled poly(l-lactide) in a shape of rose flower has been successfully polymerized on the surface of the β-tricalcium diphosphate(V) nanocrystals. The studied materials were evaluated in vitro including cytotoxicity (MTT assay) and hemolysis tests. The obtained results suggested that the studied materials may find potential application in tissue engineering.

## 1. Introduction

Theranostics is a field of modern nanomedicine that is based on targeted therapy and medical diagnostic tests. This approach is involved with a dynamic development of chemistry, biology, biotechnology, physics and last, but not least—nanotechnology [1].

In the case of biomaterial, some requirements should be met to be treated as a theranostics agent, but one is obligatorily related to similarity to the original tissue [2]. Moreover, the bone minerals are mostly formed by calcium hydroxyapatites (herein CaHAp) [3]. There are a lot of materials related to the apatite family that could be used in the theranostics applications. Among them is β-tricalcium diphosphate (herein β-TCP) that could be a promising agent. This material crystallizes in rhombohedral structure in *R3C* space group [4]. Furthermore, the CaHAp and β-TCP are biocompatible, osteoconductive, and bioactive biomaterials. Comparing the Ca:P ratios in both materials, hydroxyapatite is more similar to mineralized bone, however resorption time of the β-TCP-based materials is much shorter due to its much greater solubility in water in 37 °C [5].

The simplest way to use this material as a bio-imaging agent is a structural modification in order to observe light emission in the range of one of biological optical windows. Our research has showed that β-TCP doped with Pr^3+^ ions exhibits an intense emission band near 650 nm. It has been shown that luminous efficiency of Pr^3+^ ions could be further enhanced by Ce^3+^ co-doping. The literature reports that significant increase of Pr^3+^ emission via energy transfer from excited Ce^3+^ions is observed i.e., in YAG (Y_3_Al_5_O_12_, Yttrium Aluminum Garnet) crystals [6]. Efficient luminescent properties are needed to precisely monitor temporal stability and spatial migration of introduced material in the bone tissue.

To insert inorganic material into the bone tissue, a carrier agent is needed. In human bone tissue, calcium hydroxyapatite crystals are distributed in the collagen fibrils forming a complex structure. Another way to obtain biomaterial similar to bone tissue is to use bioresorbable polymer. One of the most used ones is polylactide (PLA).

Biodegradable aliphatic polyesters, for example polylactide (PLA), are widely used polymers in a variety of bioapplications such as controlled drug release, gene therapy, regenerative medicine, or implants [7,8,9,10,11]. The most effective and controlled method for PLA synthesis is metal-catalyzed/-initiated ring-opening polymerization (ROP) of lactide [12,13,14]. Among the wide variety of catalytic systems, the most attractive so far are the single-site initiators based on structural motif L-M-OR, where L is the ancillary ligand, M is the metal center, and OR is the initiating group [15,16,17,18,19,20,21,22,23]. Alternative, similarly attractive in the controlled synthesis of PLA are binary catalytic systems based on the homoleptic complexes and external alcohol combination [24,25,26]. The most widely used system includes commercially available bis(2-ethylhexanoate)tin(II) (Sn(Oct)_2_) catalyst, commonly applied in industry [12]. However, in the context of medical applications, biologically benign complexes (Zn, Mg, Ca) are the most searched due to their innocuous nature, ready availability, and their effectiveness for polymerization, both in terms of activity and stereoselectivity. Recently, polymer/inorganic nanocomposite materials have attracted considerable interest because of their excellent properties through synergism of polymer and inorganic nanoparticle (iNPs) components. The key to obtaining suitable in bioaplications polymer/iNPs composites is achieving the fine dispersion of inorganic nanoparticles in the polymer matrix. In terms of variety of synthetic strategies, some of them are more favorable for the formation of polymer/iNPs composites because of the simplicity in materials processing. Among others, it is worth mentioning about binding of polymer chains to iNPs by coating modification, dispersion of iNPs in polymer matrix, and formation of stabilizing polymer shell. PLA is suitable for surface coating of iNPs because of their biocompatibility and versatility while providing a platform for further biological modifications.

The proposed research has been intended to consider the well-characterized theranostics materials for bone damage tissue. Nanocrystalline β-tricalcium diphosphate (β-TCP) (so-called therapeutic part) has been co-doped with Ce^3+^ and Pr^3+^ ions (diagnostic part) and coated with poly(l-lactide) (so called a carrier). Moreover, they have been tested in vitro to examine their cytotoxicity using human chondrocyte cell line and mouse osteoblast cell line (MTT assay) and a standard hemolysis test.

## 2. Materials and Methods

### 2.1. Materials

All reactions and operations that required an inert atmosphere of N_2_ (synthesis of zinc complex L_2_Zn and polymers PLA) were performed using a glovebox (MBraun, Garching, Germany) or standard Schlenk-like apparatus (ILT&SR PAS, Wroclaw, Poland) and vacuum line techniques. The solvents for synthesis were purified by standard methods before use: n-hexane (VWR, Radnor, PA, USA) distilled from Na; MeOH (HPLC, VWR, Radnor, PA, USA) distilled from CaH_2_; CH_2_Cl_2_ (99.8% VWR, Radnor, PA, USA) distilled from P_2_O_5_; deuterated solvents (C_6_D_6_), distilled from NaH. Unless otherwise stated, all chemicals were purchased from commercial sources and used without further purification:

Calcium carbonate (99.5%, Alfa Aesar, Haverhill, MA, USA), ammonium dihydrogen phosphate (99+%, for analysis, ACROS Organics, Geel, Belgium), cerium(III) nitrate hexahydrate (99.999%, Alfa Aesar, Haverhill, MA, USA), praseodymium(III, IV) oxide (99.999%, Sigma-Aldrich, Saint Louis, MO, USA), nitric acid (65%, Suprapure, Merck, Darmstadt, Germany), citric acid (99%+, Alfa Aesar, Haverhill, MA, USA), ethylene glycol (ultrapure, Avantor Performance Materials Poland S.A., Gliwice, Poland) 2,4-di-*tert*-buthylphenol (99%, Sigma-Aldrich, Saint Louis, MO, USA), formaldehyde (37% solution in H_2_O, Sigma-Aldrich, Saint Louis, MO, USA). *N*-methylcyclohexylamine (99%, Sigma-Aldrich, Saint Louis, MO, USA), ZnEt_2_ (1.0 M solution in *n*-heptane, Sigma-Aldrich, Saint Louis, MO, USA). Proligand [*L^Cy^-H*] *N*-[methyl(2-hydroxy-3,5-di-*tert*-butylphenyl)]-*N*-methyl-*N*-cyclohexylamine was synthesized according to a literature procedure [24].

### 2.2. Synthesis of β-tricalcium diphosphates (β-TCP) Doped with Pr^3+^and Co-Doped with Ce^3+^ Ions

The nanocrystalline β-Ca_3_(PO_4_)_2_ calcium phosphate co-doped with Ce^3+^ and Pr^3+^ ions were prepared by modified Pechini’s method, using 18 mmol of CaCO_3_, 12 mmol of NH_4_H_2_PO_4_, 0.09 mmol of Ce(NO_3_)_3_·4H_2_O, and 0.015 mmol of Pr_6_O_11_. Intentional concentration of the Ce^3+^ and Pr^3+^ ions was set to 0.5 mol%, in replacement of overall molar content of Ca^2+^ ions.

In this method, stoichiometric amounts of CaCO_3_ and Pr_6_O_11_ were weighed and digested in excess of HNO_3_ (Suprapure, Merck) in order to transform them into nitrates. Subsequently, cerium nitrate was dissolved in deionized water together with calcium and praseodymium (III) nitrates. Afterwards, the excess (12,5-fold relative to the total amount of cations) of citric acid as well as ethylene glycol were added under constant stirring at 60 °C, resulting in a viscous mixture. Finally, a suitable amount of ammonium hydrogen phosphate was added. The temperature was raised up to 120 °C. The heating was continued until a white voluminous foam was obtained. The mixture was further dried for 3 days at 90 °C. Afterwards, the resin thus obtained was calcinated in a temperature of 900 °C. As a result, a white powder (β-TCP) was obtained.

### 2.3. Synthesis of poly(l-lactide) (PLLA)

Synthesis of PLLA was divided into two parts. At first, the initiator was synthetized. Afterwards, ring opening polymerization of L-lactide was carried out. Oligomeric chains based on L-lactide with methyl ending group were prepared in ring opening polymerization (ROP) reaction. Homoleptic zinc aminophenolate complex type ZnL_2_ (*L* = *N*-methyl-(2-hydroxy-3,5-di-tert-buthylphenyl)]-*N*-cyclohexylamine) as initiator was used. High efficiency of this catalyst was described in the literature [24,25,26].

#### 2.3.1. Synthesis of Initiator

[(L^Cy^)_2_Zn] was synthesized following a modified procedure of our previous report [25].

To a stirred solution of 0.66 g (2.00 mmol) of proligand L^Cy^-H in *n*-hexane (20 mL), 1.00 mL (1.00 mmol) of ZnEt_2_ (1.0 M solution in *n*-heptane) was added dropwise at ambient temperature. Next, the solution was stirred for 12 h until a crude product precipitated. The white powder of [(L^Cy^)_2_Zn] was collected by filtration, washed with cold n-hexane, and dried in vacuo. Yield: 92% (0.67 g, 0.92 mmol). Anal. Calcd (Found) for C_44_H_72_N_2_O_2_Zn (726.41): C 72.75 (72.83), H 9.99 (9.86), N 1.96 (1.85)%. ^1^H NMR for major form (500 MHz, C_6_D_6_, 298 K): δ = 7.72 (2H, d, ArH, *J*_HH_ = 2.6 Hz), 7.09 (2H, d, ArH, *J*_HH_ = 2.6 Hz), 4.16 (2H, d, N-CH_2_-Ar, *J*_HH_ = 12.1 Hz), 3.47 (2H, d, N-CH_2_-Ar, *J*_HH_ = 12.1 Hz), 3.20 (2H, m, C_6_H_11_), 2.48 (6H, s, NCH_3_), 1.74 (18H, s, C(CH_3_)_3_), 1.66 (10H, m, C_6_H_11_), 1.59 (18H, s, C(CH_3_)_3_), 1.05 (10H, m, C_6_H_11_). ^13^C NMR (500 MHz, C_6_D_6_, 298 K) δ = 164.3, 138.4, 135.5, 125.9, 124.4, 120.5 (12C, Ar), 65.3 (N-CH_2_-Ar), 61.9 (2C, C_6_H_11_), 36.8 (C(CH_3_)_3_), 35.5 (C(CH_3_)_3_), 33.8 (NCH_3_), 32.3 (C(CH)_3_)_3_), 30.2 (C(CH)_3_)_3_), 26.9, 26.1, 24.4 (10C, C_6_H_11_).

#### 2.3.2. Representative Procedure for Solution Polymerization

ROP polymerization of *l*-Lactide while using binary catalytic system [(*L^Cy^*)_2_Zn]/MeOH.

The solution of zinc complex [(*L^Cy^*)_2_Zn] in CH_2_Cl_2_ (15 mL) was placed in a Schlenk flask, and MeOH and *l*-lactide in molar ratio [(*L^Cy^*)_2_Zn]/MeOH/*l*-Lactide 1/1/15 was added. The obtained solution was stirred for 5 h. At certain time intervals, about 1 mL aliquots were removed, precipitated with hexanes, and dried in vacuo. The obtained precipitates were dissolved in C_6_D_6_ and used for the conversion monitoring, which was determined by ^1^H NMR. After the reaction was completed, an excess of hexanes was added to the reaction mixture. The obtained crude polymer was next filtered off and dried in vacuo. The resulting powder was dissolved in dichloromethane and the PLA was precipitated with excess of cold hexanes. The PLA was collected by filtration, washed with hexanes, and dried in vacuo. The reaction mixture was prepared in glovebox, and the next subsequent operations for the isolation of pure PLLA were performed by using standard Schlenk apparatus and vacuum line techniques.

### 2.4. Representative Procedure for Preparation of β-TCP@PLLA Composites

TCP@PLA composites were obtained by using precipitation method.

To the sample of polymer, PLLA (50 mg) dissolved in 1 mL of CH_2_Cl_2_ nanocrystalline β-TCP (5 mg) was added and the mixture was stirred at room temperature for 0.5 h. The β-TCP@PLLA composites were precipitated with an excess of hexanes (50 mL), which was added dropwise over 10 min. Next the hexane was removed by decantation and composite material was purified by washing in hexane (3 × 50 mL) and isolated by drying the precipitate afterwards under reduced pressure during 72 h in the temperature 60 °C.

### 2.5. Physicochemical Characterization

^1^H, ^13^C NMR spectra were obtained using Bruker Avance 500 MHz spectrometer (Bruker, Billerica, MA, USA). The chemical shifts are given in ppm and referenced to the residual protons in the deuterated solvents. Microanalyses were conducted with an Elementar CHNS Vario EL III analyzer (Elementar, Langenselbold, Germany).

X-Ray Powder Diffraction (XRD) studies were carried out using PANalytical X’Pert Pro diffractometer (Malvern Panalytical Ltd, Malvern, UK) equipped with Ni-filtered Cu Kα radiation (*V* = 40 kV, *I* = 30 mA, *λ* = 1.5406 Å). The XRD patterns were collected during 3 h in the 2*θ* range of 10–60°.

The microstructure investigations and elemental analysis were carried out using a scanning electron microscope FEI Nova NanoSEM 230 (FEI Company, Hillsboro, OR, USA) equipped with EDS spectrometer (EDAX PegasusXM4) (EDAX Inc., Mahwah, NJ, USA). TCP sample was attached to the measuring table using graphite tape and imaged at an accelerated voltage 10 kV. In order to eliminate the effect of collecting charge on the surface of material, PLLA and composite samples were imaged at an accelerated voltage 5 kV and 10 kV.

High-resolution emission spectra and luminescence kinetics curves were recorded using excitation wavelength 445 nm. Opolette Nd:YAG Laser-OPO system (Opotek INC, CA, USA) was used as an excitation source. Emission spectra were recorded with use of DongWooOptron d750 monochromator (DongWoo Optron, Maesan-ri, South Korea). Light was collected by Hamamatsu R928 photomultiplier (Hamamatsu Photonics K.K., Hamamatsu, Japan). Signal from the photomultiplier was analyzed in parallel by SR250 Gated Integrator (Stanford Research System, Sunnyvale, CA, USA) for integration of signal and a Tektronix TDS 3050 digital oscilloscope (luminescence kinetics) (Tektronix Inc., Beaverton, OR, USA).

### 2.6. Evaluation of Biological Properties

Potential non cytotoxic properties of obtained materials were evaluated via MTT assay by using human chondrocyte cell line (TC28A2) and mouse osteoblast (7F2) cell lines. Hemolysis assay by using ram blood cells was performed to estimate possible hemolytic activity of our composites.

#### 2.6.1. Human Chondrocytes Cell Line

TC28A2 human chondrocyte cell line was maintained in high glucose Dulbecco’s Modyfied Eagle Medium (DMEM) with l-glutamine (Biowest, Nuaillé, France) and supplemented with 10% Fetal Bovine Serum (FBS) South America Heat Inactivated (Biowest, Nuaillé, France), 200 U/mL penicillin, and 200 µg/mL streptomycin. Mouse osteoblast cell line (7F2) was cultured in Minimum Essential Medium Eagle–alpha modification (*α*-MEM) without nucleosides (Biowest, Nuaillé, France). To obtain a full cultured medium, *α*-MEM was supplemented with 10% FBS and 2mM stable glutamine (Biowest, Nuaillé, France). TC28A2 and 7F2 cell lines were incubated in standard conditions at 37 °C in humidified atmosphere of 5% CO_2_ and 95% air. Passive cells were used three times in the experiments.

#### 2.6.2. Influence of Obtained Composites on Chondrocytes Proliferation Rate

Proliferation capacity of human chondrocytes and mouse osteoblast were evaluated via performing MTT cytotoxicity assay Tc28A2 and 7F2 cells were seeded at density 10,000 cells per well in 96-well plates and allowed to attached and grow for 24 h. Then, cells were washed with sterile PBS (Biowest, Nuaillé, France), and fresh medium and adequate concentration of tested compounds, (50 µg/mL and 100 µg/mL of β-Ca_3_(PO_4_)_2_, PLLA and composite) was added. MTT (BioReagent, >97.5%, Sigma-Aldrich, Saint Louis, MO, USA) assay was performed 24 h after cells treatment. Treatment medium was removed and sterile PBS containing 0.5 mg/mL MTT (tiazol blue tertazolium) was added, and cells were incubated 3 h at 37 °C. After incubation, medium containing MTT was removed without washing, and formed formazan crystals were dissolved in DMSO (99.5%, Sigma-Aldrich, Saint Louis, MO, USA). Absorbance was read with a Varioskan LUX plate reader (ThermoFisher Scientific, Waltham, MA, USA) at 560 nm with background reference at 670 nm. The experiment was performed three times. Percentage of cell viability was calculated using the following formula:(1)$$\mathrm{Cells}\;\mathrm{viability}\;=\frac{\mathrm{sample}\;\mathrm{absorbance}}{\;\mathrm{control}\;\mathrm{absorbance}}\times 100\%.$$

As a reference control (100% of cells viability), samples of non-treated cells were used in both cell lines.

#### 2.6.3. Hemolysis Assay

Hemolysis assay was performed according to the protocol described elsewhere with the slight modification [27]. Ram blood (ProAnimali, Wroclaw, Poland) was centrifuged (3000 RPM, 10 min) in order to obtain erythrocyte fraction, which was washed with PBS (phosphate-buffered saline, pH 7.4) and mixed with fresh PBS (1:1 *v/v*). β-Ca_3_(PO_4_)_2_; PLLA and the β-Ca_3_(PO_4_)_2_/PLLA (1:10) composite were mixed with erythrocytes at final concentration of 50 and 100 µg/mL and incubated in 37 °C for 2 h. Then, samples were centrifuged in order to obtain supernatant (5000 RPM, 5 min) and the optical density was measured at 540 nm with a Varioskan LUX plate reader (ThermoFisher Scientific, Waltham, MA, USA). As a reference control (100% of hemolysis), the 1% solution of SDS (sodium dodecyl sulfate) was used and as negative control the solution of PBS (phosphate-buffered saline) was applied. Obtained results were compared with the absorbance of SDS sample and shown as a percentage of hemolysis and the hemolysis percentage was calculated as using following formula:(2)$$Hemolysis=\frac{sample\;absorbance\;-\;negative\;control\;absorbance}{positive\;control\;absorbance\;-\;negative\;control\;absorbance}\times 100$$

## 3. Results and Discussion

### 3.1. Structural Properties

#### 3.1.1. β-tricalcium diphosphate (β-TCP)

Structure and phase purity were checked with the powder XRD technique and were compared with the reference standard of the tetragonal β–Ca_3_(PO_4_)_2_ lattice ascribed to the *R-3c* space group. Diffractograms of β-TCP co-doped with Ce^3+^ and Pr^3+^ and TCP@PLLA composite as well as the theoretical XRD pattern of β–Ca_3_(PO_4_)_2_ (ICSD 97500) and the diffractogram of pure PLLA are shown in Figure 1.

β-TCP crystallized in rhombohedral structure, in the space group *R3c*. Due to Shannon, Ca^2+^, Ce^3+^, and Pr^3+^ effective ionic radii were 1.00 Å, 1.01 Å, and 0.99 Å, respectively [28]. Due to similar ionic radiilanthanide, ions substituted Ca^2+^ sites in the β-TCP matrix lattice.

#### 3.1.2. Poly(l-lactide) (PLLA)

The activity and effectiveness of binary catalytic system [(*L^Cy^*)_2_Zn]/MeOH to synthesis of PLLA in living ROP of lactides from high and ultra-low molecular weight was recently published. That catalytic system is well suited especially for the synthesis of precisely defined oligolactides in comparison with the commercial Sn(Oct)_2_, which is not as selective as [(*L^Cy^*)_2_Zn]/ROH. Additionally, the most popular Sn(Oct)_2_ used for the synthesis of low molecular PL*L*A produces the oligolactides with the fraction of alkyl-(*S,S*)-O-lactyllactate. Therefore, the precisely defined polymer matrix containing planned 15 lactide units has been obtained by using [(*L^Cy^*)_2_Zn]/MeOH catalytic system. Under selected molar ratio of ROP components [(*L^Cy^*)_2_Zn]/MeOH/*L*-LA = 1/1/15, the oligomer 15-PL*L*A-Me were obtained (Figure 2).

The end groups and the number of lactide units were detected by using NMR spectroscopy. The ^1^H NMR spectra for 15-PLLA-Me oligomer showed expected resonances for both chains end, methyl ester and hydroxyl groups, and oligolactide backbone chain (Figure 3). The most intense signals A_3_ and B_3_ corresponded to methine and methyl groups of repetitive central open lactide units. The adequate resonances of open lactide units close to end groups were denoted as A_1-2_ B_1-2_ (a couple of signals coming from neighboring mer to hydroxy end) and A_4-5_ B_4-5_ (the first open lactide molecule close to ester chain end) (see Figure 2).

### 3.2. Morfological Properties

SEM images of uncoated β-TCP (A) and poly(l-lactide) (B) compared to β-TCP coated with PLLA in two β-TCP:PLLA ratios: 1:10 (C) and 1:20 (D) are presented in Figure 4.

The surface of the sample coated with 1:20 ratio (D) clearly showed new morphology with the shape of rose flower, while surfaces of pure PLLA (B) and the sample coated with 1:10 ratio (C) were much smoother.

### 3.3. Spectroscopic Properties

#### 3.3.1. Emission Spectra

Emission spectra were measured in the range of wavelength from 500 nm to 700 nm after excitation with 445 nm. Emission spectra of β-TCP:0.5%Ce; 0.5%Pr sintered at 900 °C for 3 h, pure and coated with poly(l-lactide) upon excitation at 445 nm, are presented in Figure 5.

The Ce^3+^ and Pr^3+^-co-doped tricalcium β-diphosphate showed prominent ^1^D_2_ → ^3^H_4_ and ^3^P_0_ → ^3^F_2_ transitions. The ^3^P_0_ → ^3^F_2_ transition was less intense than the ^1^D_2_ → ^3^H_4_, but it was observed near 650 nm—in the range of first optical window for biological tissues. Spectrum of pure β-TCP co-doped with Ce^3+^ and Pr^3+^ ions was identical to the spectrum of coated with PLLA. This means that the presence of the coating layer did not change luminescent properties of β-TCP core. Furthermore, resemblance of both spectra suggests that Pr^3+^ ions remained in phosphate structure and did not diffuse to polymer body.

#### 3.3.2. Luminescence Kinetics

Luminescence kinetics curves were recorded after excitation with 445 nm. Decays from the *^1^D_2_* and *^3^P_0_* state were observed at subsequently 605 and 650 nm.

Lifetimes of the *^1^D_2_* and *^3^P_0_* states of Pr^3+^ ions were not mono-exponential. Average lifetimes of all materials were calculated from the equation (Table 1):(3)$$\tau_{avg}=\frac{\int I\left(t\right)tdt}{\int I\left(t\right)dt}$$
where: *I*—intensity; *t*—time; *τ_avg_*—average lifetime.

*^1^D_2_* lifetimes in pure TCP and 10% composite with PLLA were the same, but in the 5% composite, they were a little bit shorter; however, differences between samples were too scarce to consider them as a significant influence of polymer coating. Decay curves observed at 650 nm and 605 nm are presented in Figure 6.

Decay time of the luminescence from the state *^3^P_0_* observed at 650 nm was about 1 µs in each studied case, which was much longer than the luminescence lifetime of emission exhibited by organic molecules from biological tissue [29,30]. This feature could be used to easily discern whether observed luminescence comes from tissue-related molecules or artificially introduced β-TCP. Time dependency and spectral shape of luminesce from introduced material could be used to precisely monitor temporal stability and spatial migration of β-TCP in the bone tissue. Location and stability monitoring are needed to fulfil diagnostic part of β-TCP/PLLA composite possible theranostics application.

### 3.4. Biological Features

#### 3.4.1. Cytotoxicity Assay

Our results clearly showed that some of the tested materials increased cell proliferation rate in both concentrations 50 µg/mL and 100 µg/mL in the TC28A2 cell line, which may indicate enhancement of metabolic activity of treated cells. PLLA seems to be degraded in an extracellular environment to l-lactide and then may be transported via MTCs (monocarboxylate transporters) and used by chondrocytes as an energetic fuel [31]. Additionally, chondrocytes proliferation rate was higher than 50% even in the 100 µg/mL concentration of all tested composites. Obtained results were collected in Figure 7. Intriguingly, in comparison with chondrocytes, viability of 7f2 cells treated with PLLA maintain edat 74% in concentration 50 µg/mL and seemed to gradually decrease in higher concentrations. The lower proliferation rate of 7F2 cells may be explained by increased lactate production in osteoblasts, which was caused by the aerobic glycolysis even in the presence of oxygen. This type of metabolism was similar to the Warburg effect, the major hallmark of cancer. It transpired that specific metabolic needs of osteoblasts demanded this particular type of metabolism. Elevated concentrations of lactic acid may additionally acidify extracellular environment, thus osteoblast cells do not use PLLA as an energy fuel like chondrocytes [32]. When compared to other compounds, the viability of mouse osteoblasts maintains around 75% when treated with β-TCP and the composite at a concentration of 50 µg/mL and around 60% at a concentration of 100 µg/mL. This may indicate that the osteoblast cell line is more sensitive to alterations in extracellular environment. However, what is the most important is that proliferation of TC28A2 and 7F2 cell lines was maintained above 50% in each sample and both concentrations.

#### 3.4.2. Hemolysis

Tested compounds at concentrations of 50 and 100 µg/mL did not cause the hemolysis of ram erythrocytes. Obtained results were compared with the hemolysis caused by 1% of SDS solution (complete damage of the cellular membrane and hemoglobin release; data not shown) and are presented in Figure 8. Statistical analysis was also performed (*p* < 0.05) using one-way ANOVA test.

Performing such an experiment is essential for chemical compounds designed to be applied in direct contact with the human body and its fluids. The acceptable range of hemolysis is set below 5% [33], therefore tested substances could be regarded as erythrocyte safe. Similar results were obtained by Zibiao et al. who proved that selected polymers based on PLLA and PEG did not cause an hemolysis higher than 5%, even in higher concentrations [34]. The researchers proved that coating some commercial metal alloys with PEO/PLLA (plasma electrolytic oxidized/poly(l-lactide) composites can help reduce their hemolytic potential [35]. β-TCP was also tested before in order to check its hemolytic potential, but such effect was not observed [36].

## 4. Conclusions

In this work, the novel and self-assembled poly(l-lactide) (PLLA)/β-tricalcium diphosphate (β-TCP) composite has been prepared. In the first step, the well-β-TCP co-doped with Ce^3+^ and Pr^3+^ ions was obtained. Further, it was prepared precisely tailored to low-molecular mass PLLA on the β-TCP surface using biocatalysts from the group of zinc aminophenolates. The obtained materials may be potentially used to bone grafting and in vivo bioimaging in the near-infrared (NIR) window (also known as “optical window” or “therapeutic window”). PLLA has been used to enhance biocompatibility of the obtained composite for further application. Moreover, the polymer did not affect the luminescent properties of Ce^3+^ and Pr^3+^ ions co-doped β-TCP.

Furthermore, the hemolysis higher than 5% was not observed for all studied materials. It has been suggested that the obtained materials could be safely used in vivo because the cell proliferation and mitochondrial activity have not been disrupted. In conclusion, it might not interrupt the OXPHOS (Oxidative phosphorylation) and energy production in chondrocytes.

## Figures and Tables

**Figure 1 nanomaterials-10-00331-f001:**
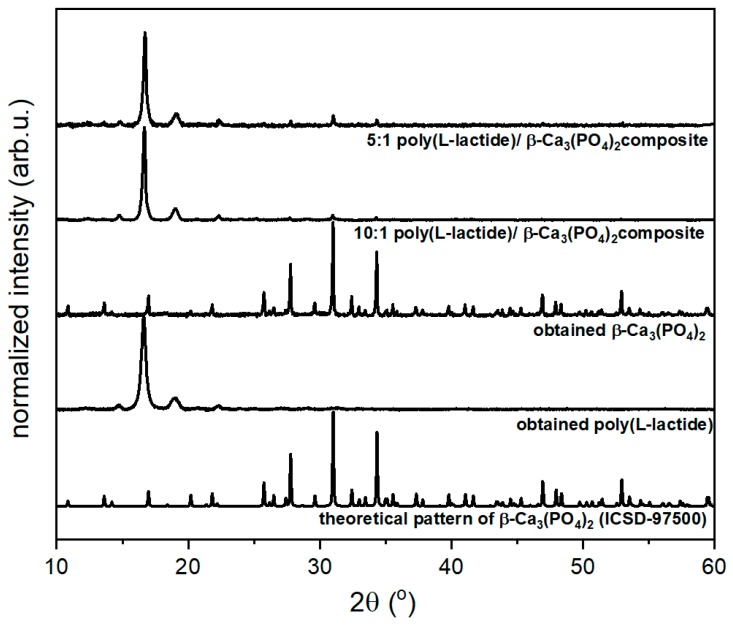
X-ray diffraction patters of poly(l-lactide) (PLLA), β-Ca_3_(PO_4_)_2_ (β-TCP) co-doped with Ce^3+^ and Pr^3+^ ions and PLLA with β-Ca_3_(PO_4_)_2_ composite compared to theoretical pattern of β-Ca_3_(PO_4_)_2_ (ICSD-97500).

**Figure 2 nanomaterials-10-00331-f002:**
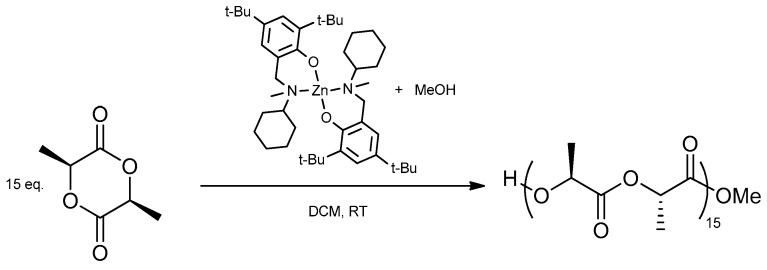
Synthesis of 15-PL*L*A-Me oligomer.

**Figure 3 nanomaterials-10-00331-f003:**
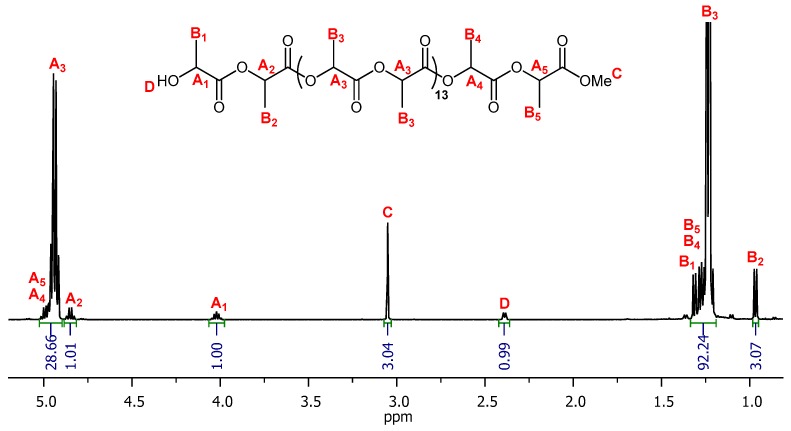
^1^H NMR spectrum of 15-PL*L*A-Me oligomer (*C_6_D_6_*).

**Figure 4 nanomaterials-10-00331-f004:**
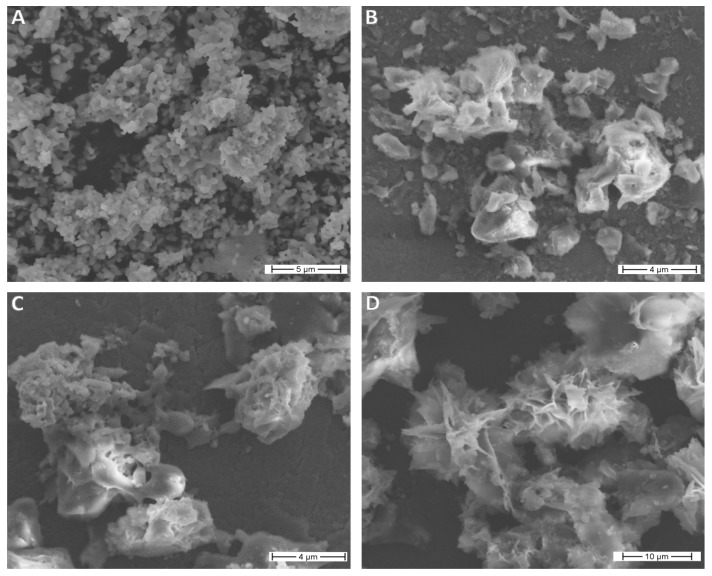
SEM images of (**A**) β-TCPco-doped with Ce^3+^ and Pr^3+^ ions, (**B**) PLLA, (**C**) PLLA with β-TCP composite with phosphate:polymer 1:10 ratio, and (**D**) PLLA with β-TCP composite with phosphate:polymer 1:20 ratio.

**Figure 5 nanomaterials-10-00331-f005:**
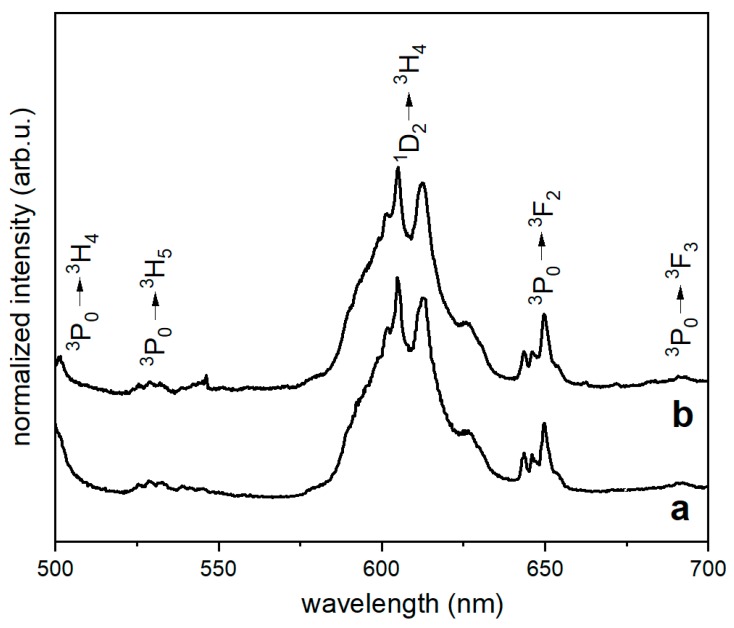
Emission spectra of PLLA with β-TCP co-doped with Ce^3+^ and Pr^3+^ ions (**a**) and PLLA with β-TCP composite with phosphate:polymer 1:20 ratio (**b**). Spectra were recorded after excitation with 445 nm.

**Figure 6 nanomaterials-10-00331-f006:**
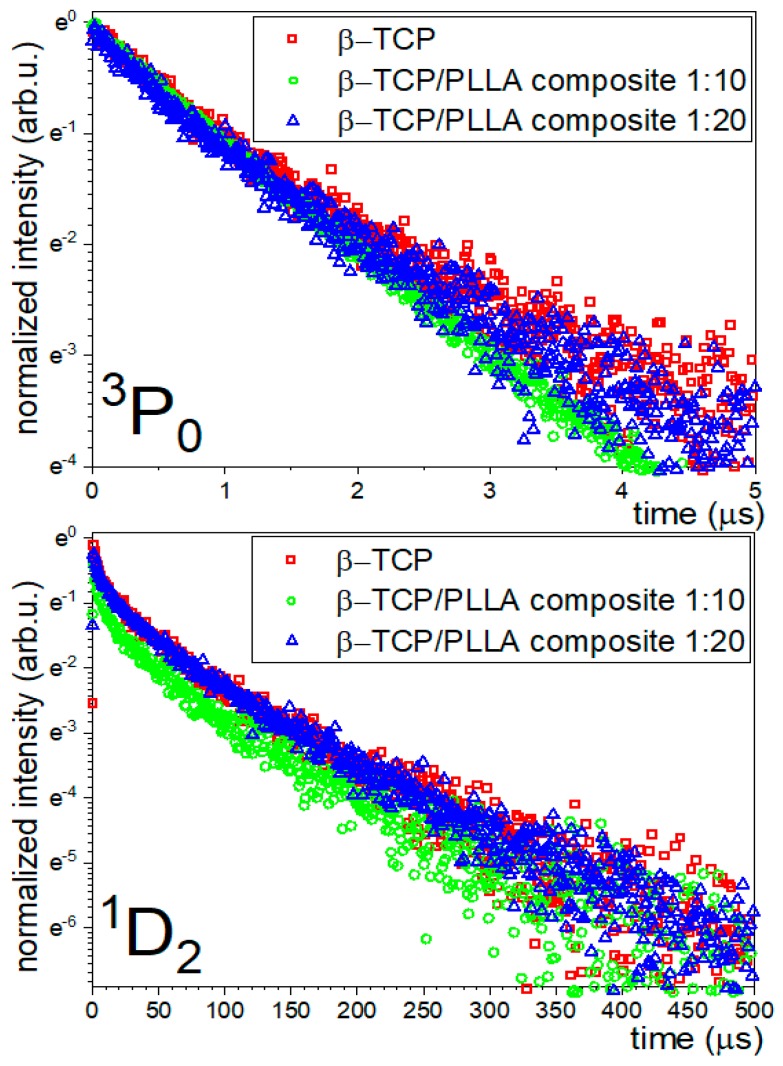
Luminescence kinetics curves of β-TCP co-doped with Ce^3+^ and Pr^3+^ ions PLLA with β-TCP composite with phosphate: polymer 1:10 ratio and PLLA with β-TCP composite with phosphate: polymer 1:20 ratio, recorded at 650 nm (*^3^**P_0_*) and recorded at 605 nm (*^1^**D_2_*).

**Figure 7 nanomaterials-10-00331-f007:**
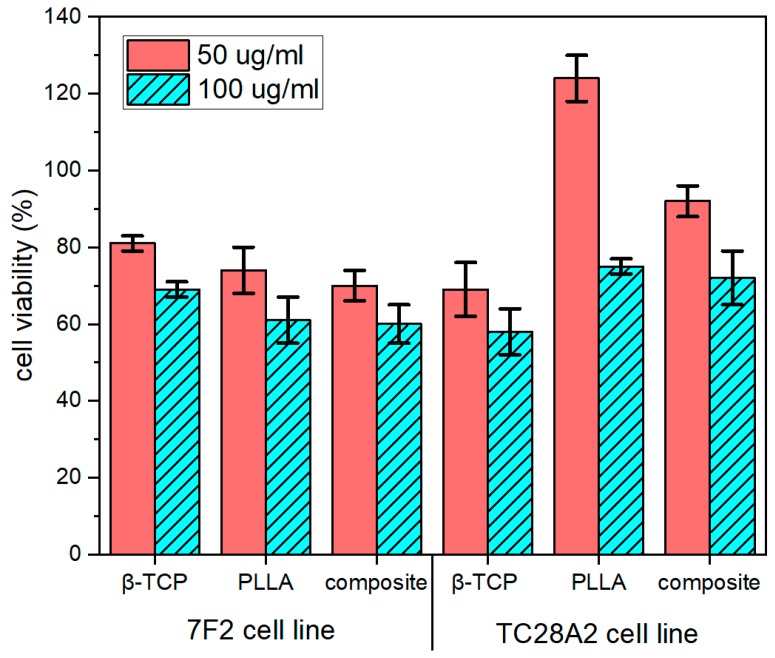
7F2 and TC26A2 cells viability after 24 h of incubation and exposition on β-TCP co-doped with Ce^3+^ and Pr^3+^ ions, PLLA, and PLLA with β-TCP composite with phosphate:polymer 1:20 ratio, in concentrations of 50 µg/mL and 100 µg/mL.

**Figure 8 nanomaterials-10-00331-f008:**
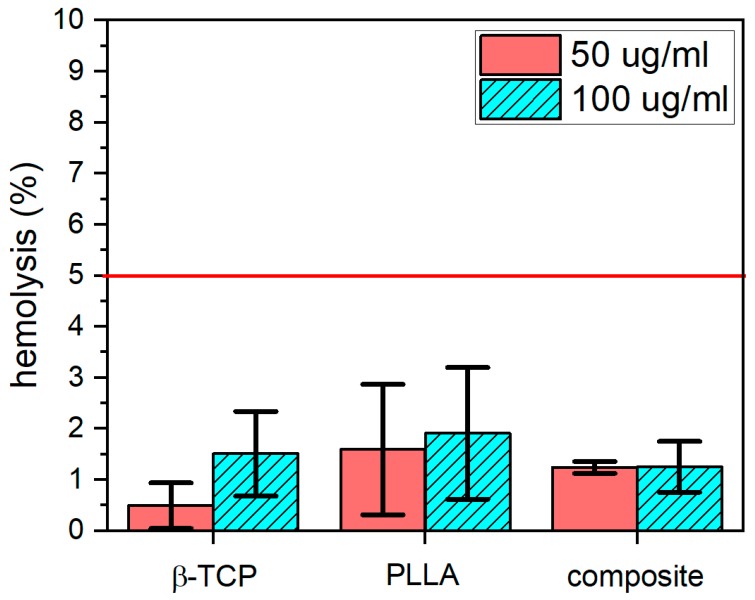
Activity of β-TCP co-doped with Ce^3+^ and Pr^3+^ ions, PLLA, and PLLA with β-TCP composite with phosphate:polymer 1:20 ratio, in concentrations 50 µg/mL and 100 µg/mL in reference to 1% of SDS solution (mean ± SD, *n* = 3; all results were statistically significant). The red line indicates the acceptable hemolysis level.

**Table 1 nanomaterials-10-00331-t001:** Calculated lifetimes of β-TCP co-doped with Ce^3+^ and Pr^3+^ ions PLLA with β-Ca_3_(PO_4_)_2_ composite with phosphate:polymer 1:10 ratio and PLLA with β-TCP composite with phosphate:polymer 1:20 ratio. Delays from the state ^3^P_0_ were observed at 650 nm and delays from the state ^1^D_2_ were observed at 605 nm.

Materials	State	Observed Wavelength [nm]	Lifetime [µs] (*τ_avg_*)
β-TCP:0.5%Ce/0.5% Pr, 900 °C	*^3^P_0_*	650	1.31
	*^1^D_2_*	605	84.39
β-TCP/PLLA composite 1:20	*^3^P_0_*	650	1.28
	*^1^D_2_*	605	84.50
β-TCP/PLLA composite 1:10	*^3^P_0_*	650	1.34
	*^1^D_2_*	605	77.28

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
