# Peer review of "Preparation and Characterization of Self-Assembled Poly(l-Lactide) on the Surface of β-Tricalcium Diphosphate(V) for Bone Tissue Theranostics"

_nanomaterials, 2020, doi:10.3390/nano10020331_

Round 1

Reviewer 1 Report

The proposed manuscript from Jan Zienkiewicz et al. describes the development of  biocomposites based on β-tricalcium diphosphate(V) co- doped with Ce3+ and Pr3+ modified by poly(L-lactide) for bone  tissue theranostics. The composites were spectroscopically characterized and their morphological and structural properties were studied. Finally cytotoxicity and hemolysis test were performed.

After the revisions the proposed work is much more clear. The introduction as well as the goal of the research better explained and the experimental part more detailed. The English has been improved despite fine/minor spell check isrequired.

I feel that this paper could be published after minor revisions:

In the title, “Preparation and characterization self-assembled” should be replaced with “Preparation and characterization of self-assembled” The abstract should be revised adding the aim and the conclusion of the work In line 312 please, replace “Biological featurest” with “Biological features”

Author Response

Dear Editor,

We would like to express our sincerest gratitude to the Reviewers for their enormous efforts in criticizing the manuscript. We have taken into account all raised question here follows the detailed answers to the Reviewers. Moreover, all changes we have made to the original manuscript, are marked in the red color in the text.

Reviewer:

Comments and Suggestions for Authors

The proposed manuscript from Jan Zienkiewicz et al. describes the development of  biocomposites based on β-tricalcium diphosphate(V) co- doped with Ce3+ and Pr3+ modified by poly(L-lactide) for bone  tissue theranostics. The composites were spectroscopically characterized and their morphological and structural properties were studied. Finally cytotoxicity and hemolysis test were performed.

After the revisions the proposed work is much more clear. The introduction as well as the goal of the research better explained and the experimental part more detailed. The English has been improved despite fine/minor spell check is required.

 I feel that this paper could be published after minor revisions:

In the title, “Preparation and characterization self-assembled” should be replaced with “Preparation and characterization of self-assembled”. The abstract should be revised adding the aim and the conclusion of the work. In line 312 please, replace “Biological featurest” with “Biological features”

Answers:

The manuscript has been carefully reviewed and corrected in accordance with Reviewer’s suggestions.

Reviewer 2 Report

On  the basis of the changes which were made by the authors, I suggest the manuscript to be accepted as it is. 

Author Response

Dear Editor,

We would like to express our sincerest gratitude to the Reviewers for their enormous efforts in criticizing the manuscript. We have taken into account all raised question here follows the detailed answers to the Reviewers.

Reviewer 2:

Comments and Suggestions for Authors

On  the basis of the changes which were made by the authors, I suggest the manuscript to be accepted as it is.

Answers:

Thank you very much for reviewing of our article.
